# Light-Responsive Polymeric Micellar Nanoparticles with Enhanced Formulation Stability

**DOI:** 10.3390/polym13030377

**Published:** 2021-01-26

**Authors:** Kyoung Nan Kim, Keun Sang Oh, Jiwook Shim, Isabel R. Schlaepfer, Sana D. Karam, Jung-Jae Lee

**Affiliations:** 1Department of Chemistry, University of Colorado Denver, Denver, CO 80204, USA; kyoung.kim@ucdenver.edu; 2TGel Bio, Co, LTD, Seoul 06185, Korea; oks5486@nate.com; 3Department of Biomedical Engineering, Rowan University, Glassboro, NJ 08028, USA; shimj@rowan.edu; 4Division of Medical Oncology, Anschutz Medical Campus, University of Colorado, Aurora, CO 80045, USA; isabel.schlaepfer@cuanschutz.edu; 5Department of Radiation Oncology, Anschutz Medical Campus, University of Colorado, Aurora, CO 80045, USA; sana.karam@cuanschutz.edu; 6Department of Bioengineering, University of Colorado Denver, Denver, CO 80204, USA

**Keywords:** stimuli-responsive drug delivery, light-sensitive polymer, controlled release drug delivery, polymeric micellar nanoparticles

## Abstract

Light-sensitive polymeric micelles have recently emerged as promising drug delivery systems for spatiotemporally controlled release of payload at target sites. Here, we developed diazonaphthoquinone (DNQ)-conjugated micellar nanoparticles that showed a change in polarity of the micellar core from hydrophobic to hydrophilic under UV light, releasing the encapsulated anti-cancer drug, doxetaxel (DTX). The micelles exhibited a low critical micelle concentration and high stability in the presence of bovine serum albumin (BSA) solution due to the hydrophobic and π–π stacking interactions in the micellar core. Cell studies showed enhanced cytotoxicity of DTX-loaded micellar nanoparticles upon irradiation. The enhanced stability would increase the circulation time of the micellar nanoparticles in blood, and enhance the therapeutic effectiveness for cancer therapy.

## 1. Introduction

Polymeric micelles have emerged as promising drug delivery systems for cancer treatment because amphiphilic block copolymers enable micelles to improve drug solubility, control drug release, prolong circulation time in body and avoid the elimination of reticuloendothelial system (RES) [1,2,3,4,5,6,7,8]. Additionally, the prolonged circulation time makes polymeric micellar nanoparticles enhanced accumulation in solid tumors tissue with leaky vasculatures through the enhanced permeability and retention (EPR) effect [9,10,11,12]. Recently, stimulus-responsive polymeric micelles activated by pH, temperatures, redox, ultrasound, and light have been explored to provide spatial or temporal control of release of anti-cancer drugs to tumors [13,14,15,16,17,18]. Particularly, light-responsive drug delivery systems are attractive because they allow site-specific release of payload at desired disease sites with precisely controlled wavelength and intensity [19,20,21,22,23,24]. Light is already used in clinical practice in phototherapy (e.g., photodynamic and photothermal therapies) and optical imaging (e.g., fluorescence imaging and fluorescence guided surgery) [25,26].

In many light-responsive polymeric micelle systems, UV light triggers a change in polarity or a transition from hydrophobicity to hydrophilicity in domains bearing photochromic azobenzene, spiropyran, and 2-diazo-1,2-naphthoquinone (DNQ) [27,28,29,30,31,32,33], which leads to drug release [34,35,36,37,38]. However, conventional micelle systems have limited drug loading capacity (LC) and relatively high critical micelle concentrations (CMC), which impair particle stability resulting in decreased blood circulation time and release of drugs before the micelles accumulate in the tumor sites. To increase drug loading and lower CMC (i.e., to enhance the stability of micelles), micelles have been recently developed by supramolecular interactions such as π–π stacking, hydrogen bonding, stereocomplexation, and electrostatic interaction [39,40,41,42,43,44,45,46,47,48].

Here, we have developed a light-responsive polymeric micellar nanoparticle with high LC and stability using particularly π–π stacking. The light-responsiveness is imparted by poly(ethylene glycol)-block-poly-l-lysine (PEG-PLL) modified by the light-responsive DNQ group. Upon irradiation by UV or two-photon near infrared (NIR) light, hydrophobic DNQ can be converted to hydrophilic 3-indenecarboxylic acid by the Wolff rearrangement (Appendix A). Previously, photo-triggered dye release from micelles comprising a linear amphiphile, DNQ-terminated oligomer, has been reported [36]. However, these micelles had a high CMC (i.e., low stability) of 1.15 mg/mL in phosphate buffered saline (PBS). Alternative DNQ-based drug delivery systems have been developed to lower CMC, reduce the cytotoxicity, and increase biocompatibility, such as linear-dendritic PEO-b-G4 polyester conjugated with 16 DNQs [37], DNQ-modified dextran [34], and Janus-type dendritic polyamidoamine (PAMAM) amphiphiles with DNQ [35]. For in vivo drug delivery, there are still needs to enhance stability of polymeric micelle formulations, increase the drug LC, and develop new stimuli-responsive polymeric nano-sized particles for effective passive tumor targeting by the EPR effect [2,3,5]. Herein, we report a light-sensitive polymeric micellar nanoparticle activated under low power density of UV light with enhanced stability, LC, and drug retention by supramolecular hydrophobic and π–π stacking interactions between the copolymers and drugs. The micellar nanoparticles were formulated from the synthesized DNQs-conjugated PEG-PLL amphiphilic copolymers and we successfully obtained burst release of aromatic anticancer drugs, docetaxel (DTX), from the micellar nanoparticles under UV light irradiation.

## 2. Materials and Methods

### 2.1. Materials

Unless specified, chemicals were purchased from Sigma-Aldrich (St. Louis, MO, USA) and used without further purification. Methoxy-poly(ethylene glycol)-block-poly(l-lysine trifluoroacetate) (mPEG5K-b-PLKF10: Mw = 7400, 5k PEG conjugated with 10 lysine units, dispersity (Đ): 1.01–1.2) was purchased from Alamanda Polymers, Inc. (Huntsville, AL, USA) and 2-Diazo-1,2-naphthoquinone-5-sulfonyl chloride was purchased from ChemPacific (Baltimore, MD, USA). Docetaxel (DTX) was purchased from LC Laboratories (Woburn, MA, USA) and stored at −20 °C prior to use.

### 2.2. Characterizations

High performance liquid chromatography (HPLC) analysis was performed on a Hewlett Packard/Agilent series 1100 (Agilent, Santa Clara, CA, USA) equipped with an analytical C18 reverse phase column (Kinetex, 75 × 4.6 mm, 2.6 μm, Phenomenex, Torrance, CA, USA). Nuclear magnetic resonance (NMR) studies were performed on a Varian 500 system (400 MHz). The sizes and polydispersity of micelles were determined on a Delsa Nano C particle analyzer (Beckman Coulter, CA, USA). Fluorescence microscopy was conducted on a FSX 100 (Olympus, PA, USA). 8453 UV–Vis spectrophotometer (Agilent, CA, USA) was used to acquire UV–Vis spectra and Cary Eclipse fluorescence spectrophotometer was used to measure fluorescence. For transmission electron microscopy (TEM) image, a 2.0 μL aliquot of the micelle solution was deposited on a copper grid coated with a carbon film. The sample was dried at room temperature and then imaged on a Tecnai G2 Spirit BioTWIN transmission electron microscope, operating at 100 kV (note that this drying procedure employed in TEM analysis may cause considerable changes in the size and shape of the micelles since they only exist in solution).

### 2.3. Statistical Analysis

Due to the asymmetric distribution of some of our data, we did not assume a normal distribution, and data were analyzed by the non-parametric Mann–Whitney U-test. A *p* value < 0.05 was considered statistically significant unless stated otherwise. For multiple comparisons, Kruskal–Wallis tests were performed followed by Bonferroni corrections.

### 2.4. Synthesis of PEG-PLL-DNQ and PEG-PLL-Oct

Methoxy-poly(ethylene glycol)-block-poly(l-lysine trifluoroacetate) (PEG-PLL, 0.5 g, 0.068 mmol) and 2-diazo-1,2-naphthoquinone-5-sulfonyl chloride (sc-DNQ, 0.25 g, 0.93 mmol) or octanoyl chloride (c-Oct, 0.15 g, 0.93 mmol) were placed in a 100 mL round bottom flask and suspended in a CHCl_3_ (20 mL). Then, triethylamine (0.2 mL) was added slowly and the reaction mixture was stirred in the dark overnight. The solvent was evaporated under reduced pressure and the crude product was purified by size exclusion chromatography (SEC) with DMF. After evaporation of DMF under reduced pressure at 60 °C, PEG-PLL-DNQ or PEG-PLL-Oct was precipitated with ether/methanol and then centrifuged. The solvent was completely removed under high vacuum.

### 2.5. Formulation of the Micellar Nanoparticles with DTX or Nile Red (NR)

The thin-film hydration method was used to form drug-encapsulating micellar nanoparticles. Both PEG-PLL-DNQ and DTX (20% *w/w*) or NR (1% *w/w*) were dissolved in CHCl_3_ and mixed for 1 h. The organic solvent was then slowly removed using a rotary vacuum evaporation and further dried under a constant flow of nitrogen gas until a film formed in a round-bottomed flask. All organic solvents were removed in a high vacuum. After the drug-containing block copolymer film was formed, aqueous solution was added, heated up to 60 °C, and stirred to get a preliminary micellar nanoparticle solution. Subsequently, the solution was dialyzed in water for 24 h (molecular weight cutoff: 3.5 kDa). The final micellar nanoparticles were obtained by filtering through 0.22-μm filters. Similar procedures were used to formulate PEG-PLL-Oct-based micellar nanoparticles.

### 2.6. Determination of the Critical Micelle Concentration (CMC)

The pyrene fluorescence method was used to determine CMC of DNQ-conjugated micellar nanoparticles (**NP_DNQ_**) and Oct-conjugated micellar nanoparticles (**NP_Oct_**). For the fluorescence measurements, 3.0 mL of solution of **NP_DNQ_** or **NP_Oct_** containing 2.0 μM of pyrene was placed in a fluorescence quartz cuvette cell and the fluorescence emission spectra were measured in the range from 360 to 410 nm at 339 nm excitation. The concentrations of PEG-PLL-DNQ and PEG-PLL-Oct were varied from 10^−4^ to 1.0 mg/mL. The fluorescent intensities were recorded at the wavelengths corresponding to the first and third vibronic bands located at 373 nm and 383 nm. All fluorescence measurements were carried out at 25 °C.

### 2.7. Effect of Lyophilization/Reconstitution on Particle Size 

1.0 mL of DTX-loaded **NP_DNQ_** (**DTX-NP_DNQ_**, 0.5 mg/mL) was frozen and lyophilized to obtain white powder and then the resulting white powder was reconstituted with 1.0 mL of distilled water followed by addition of a concentrated PBS solution (10×). Particle sizes of **DTX-NP_DNQ_** before and after lyophilization/reconstitution were measured by DLS. The reconstituted micellar nanoparticles were used for all data. 

### 2.8. Stability Studies of NPs in the Presence of BSA

Nile Red-loaded **NP_DNQ_** (**NR-NP_DNQ_**) and **NP_Oct_** (**NR-NP_Oct_**) were incubated in bovine serum albumin solution (BSA, 10%) at 37 °C. At each time point, the micelle suspension was placed on a shaker with gentle agitation at room temperature, and then their emission spectra were recorded (excitation at 570 nm).

### 2.9. Irradiation of UV Light

To investigate the release of encapsulated DTX from **DTX-NP_DNQ_**, UV light at variable time periods was carried using a Mightex FCS fiber-coupled LED light (Mightex, Pleasanton, CA, USA) with the wavelength of 365 nm (50 mW/cm^2^).

### 2.10. Determination of DTX in **DTX-NP_DNQ_**

To determine the amount of DTX in **DTX-NP_DNQ_**, a PBS suspension of **DTX-NP_DNQ_** was place into Slide-A-Lyzer dialysis tubes (0.5 mL each tube) with a molecular weight cutoff at 3.5 kDa (Pierce, Rockford, IL, USA). These microtubes were individually dialyzed in 2.0 L of PBS buffer (1×). **DTX-NP_DNQ_** solutions from microtubes were collected separately, acetonitrile was added to dissolve the micelles, and DTX in each tube was measured by HPLC: Chromatography was performed in the isocratic mode. Mobile phase, eluent A: 0.1% orthophosphoric acid, 40%; eluent B: acetonitrile, 60%; the flow rate, 1 mL/min; the sample volume, 100 μL. The detection was carried out at 232 nm [49,50].

### 2.11. In Vitro Cytotoxicity

To compare cell proliferation by DTX, **NP_DNQ_**, and **DTX-NP_DNQ_** in normal cells (HUVECs) and breast cancer cells (MCF-7 cancer cells), HUVECs and MCF-7 cells were evaluated by (3-(4,5-dimethylthiazol-2-yl)-2,5-diphenyltetrazolium bromide) (MTT) based assay. 1 × 10^4^ cells per well were seeded into a 96-well plate and incubated at 37 °C in a humidified atmosphere with 5% CO_2_ for 24 h. Cell culture medium of HUVECs and MCF-7 cancer cells was used with EG2 media (Lonza, NJ, USA) with supplied various growth factors and RPMI 1640 supplemented with 10% FBS and 1% pen-strip, respectively. Each cell was treated with 200 μL of serially diluted samples with different concentrations (5-400 ng/mL) at 37 °C for 24 h. The negative control was treated with only cell culture media, and the empty NPs (**NP_DNQ_**) were prepared. After 24 h, they were rinsed twice with DPBS (pH 7.4, with Ca^2+^, Mg^2+^) to eliminate the remaining samples. 25 μL of the MTT reagent (5 mg/mL in media) was then added and incubated further for 4 h at 37 °C [51,52]. Thereafter, the purple precipitates (formazan crystals) were added and dissolved by adding 200 μL of DMSO. Absorbance at 570 nm was measured with a microplate reader (Gen5, Biotek, Winooski, VT, USA). The percentages of viable cells were determined through the reduction of MTT relative to the negative control.

### 2.12. Cytotoxicity of Light-Triggered **DTX-NP_DNQ_**

MCF-7 cells were grown six-well plates and various concentrations (e.g., 100, 200, and 400 ng/mL of DTX; the concentrations of NPs were adjusted to the concentration of DTX by UV-Vis spectroscopy) of **DTX-NP_DNQ_** were added. The cells were incubated for 2 h at 37 °C and then washed with fresh media (2×). Subsequently, UV light for 10 min was applied and the cells were further incubated for 24 h. The cytotoxicity of cells was determined by MTT assay.

### 2.13. Intracellular Localization of the **NR-NP_DNQ_**

To observe cellular behavior of **DTX-NP_DNQ_** in MCF-7 cancer cells, NR (hydrophobic fluorescence dye) loaded the core in **NP_DNQ_**. MCF-7 cancer cells were seeded on a 35 mm petri-dish and allowed to grow until a confluence of 60–80%. Then, they were washed twice with DPBS and the cells were incubated with 2 mL of **NR-NP_DNQ_** for up to 120 min at 37 °C. After predetermined incubation times (0, 30, 60, 90, and 120 min), the treated cells were rinsed twice with PBS and fixed by 4% paraformaldehyde for 2 h. Thereafter, 4, 6-diamidino-2-phenylindole (DAPI) (1 mg/mL in DI water) treated for nuclei staining for 3 min, and again washed twice with PBS. Fluorescent distribution in the cells was observed by a fluorescence microscope (Olympus FSX100, Olympus, Japan).

## 3. Results

### 3.1. Synthesis of the Light-Sensitive Block Copolymers and Formulation of the Micellar Nanoparticles

The synthetic route of the amphiphilic block copolymers, PEG-PLL-DNQ and PEG-PLL-Oct (Figure 1), is shown in Appendix A. PEG-PLL (*M*_n_ ~ 7400, 5k PEG conjugated with 10 lysine units) was simply coupled to DNQ-sulfonyl chloride (sc-DNQ) under a base condition to afford PEG-PLL-DNQ in 80% yield. The successful synthesis of PEG-PLL-DNQ was characterized by ^1^H NMR spectroscopy. As shown in Appendix A, the characteristic resonance signals of DNQ at 8.70, 8.47, 7.70, 7.60, and 5.29 ppm were observed in the spectrum of PEG-PLL-DNQ in DMSO-*d*_6_, which is evidence of the successful conjugation of DNQ groups on PEG-PLL. Octanoyl chloride (c-Oct) without aromatic groups was also conjugated to PEG-PLL in the same manner to form PEG-PLL-Oct that only provides hydrophobic interactions in the core of the micelle without π–π stacking interactions.

As previously reported, AB-type amphiphilic block copolymers composed of hydrophilic and hydrophobic components can self-assemble into micelles, where the hydrophobic core of the micelle can encapsulate hydrophobic drugs which will otherwise precipitate in aqueous solutions. In this study, we synthesized a new block copolymer, PEG-PLL-DNQ, containing hydrophobic light-sensitive DNQs which could generate additional π–π stacking interactions between the neighboring DNQs on the polymer backbone inside of the micelle core. Furthermore, these DNQs could also interact with DTX, which contains two aromatic phenyl groups that can form another π–π stacking in addition to hydrophobic interactions (Appendix A). These interactions will contribute to the higher formulation stability of the micelle as well as its increased loading capacity (LC) with aromatic drugs.

The DTX-loaded polymeric micellar nanoparticle (**DTX-NP_DNQ_**) was prepared by the thin-film hydration method. Dynamic light scattering (DLS) results showed that PEG-PLL-DNQ formed micelles with a volume average hydrodynamic diameter of 143 nm and TEM image revealed that **DTX-NP_DNQ_** were formed with a spherical morphology (Figure 2). For controlled experiments, **DTX-NP_Oct_** was successfully formulated with 105 nm in diameter (Appendix A).

### 3.2. Measurements of CMC, LE and LC, and Stability

CMC is a key parameter to describe the self-assembly behavior of micelles. The CMC of **NP_DNQ_** was calculated by the pyrene fluorescence method [53]. Pyrene has a fluorescence profile that is dependent on its microenvironment. In hydrophobic environments, the fluorescent intensity ratio of the first (373 nm) to third (383 nm) emission peaks (Figure 3a), the *I*_373_/*I*_383_ ratio known as polarity parameter, is low (close to 1); whereas in hydrophilic environments, the *I*_373_/*I*_383_ ratio increases (close to 2). Using this method, we were able to determine whether pyrene was in the core of the micelle (hydrophobic environments) or in aqueous solution (hydrophilic environments). The polymer concentration at which the location of pyrene changed from solution to micelle is the CMC. Figure 3a shows the fluorescence emission spectra of pyrene encapsulated in **NP_DNQ_** and the change of the *I*_373_/*I*_383_ ratio at the various concentrations of PEG-PLL-DNQ at room temperature. The curve was almost flat at low polymer concentrations but rapidly decreased at higher polymer concentrations and eventually reached a plateau. The CMC value of **NP_DNQ_** was determined by the intersection of two straight lines and the CMC was about 7.1 μg/mL (Figure 3b), which is a relevantly low CMC compared to other reported amphiphilic diblock copolymers (>20 μg/mL) and four times lower than the control micelle system, **NP_Oct_** (25 μg/mL, Figure 3c) due to π–π stacking interactions between neighboring DNQs on the polymer backbone leading to a more condensed core of **NP_DNQ_**. The reduced CMC could contribute to improved thermodynamic stability of the micelle, reduced toxicity toward cells, and increased circulating time in blood.

In addition to CMC, we evaluated the drug loading efficiency (LE) and capacity (LC) of **DTX-NP_DNQ_** and **DTX-NP_Oct_** using HPLC (Appendix A). The results show LCs of 13.5% and 7.9% for **DTX-NP_DNQ_** and **DTX-NP_Oct_**, respectively. These were associated with LEs of 90.3% and 79.1%, respectively. **NP_DNQ_** encapsulated approximately two times more DTX than **NP_Oct_**, which can be presumably ascribed that **NP_DNQ_** has a better compatibility of DTX due to π–π stacking interactions between DNQs and aromatic DTX.

Subsequently, we measured the stability of the micelles in blood plasma at 37 °C because the stability is greatly influenced by serum proteins. The hydrophobic Nile red dye (NR)-loaded micelles, **NR-NP_DNQ_** and **NR-NP_Oct_**, were incubated in 10% bovine serum albumin (BSA) solution and the intensity of fluorescence emission was monitored. NR is a fluorescent indicator for the micropolarity; it shows strong fluorescence emission intensity in hydrophobic environments while it is quenched in a hydrophilic solution [54]. As shown in Figure 3d, **NR-NP_DNQ_** was stable in BSA solution retaining their intensity up to 2 days at 37 °C (>85%) while **NR-NP_Oct_** showed a dramatic decrease in intensity, indicating release of NR into hydrophilic environments (Appendix A). These results indicate that the presence of **DNQ** on the polymer backbone increased the stability of the micelles.

### 3.3. Cellular Uptake Study and Cytotoxicity

The cell internalization of **NR-NP_DNQ_** was characterized by a fluorescence microscopy. As shown in Figure 4, **NR-NP_DNQ_** quickly entered into MCF-7 cells. After incubation with **NR-NP_DNQ_** for 30 min, most of MCF-7 cells showed red fluorescence in the cell and then the fluorescence intensity from the cells was saturated after 1 h. As shown in Figure 3d, less than 7% of hydrophobic NR was released from the micelle within 6 h. Thus, the saturation of fluorescence intensity after 1 h indicates that the cell uptake of **NR-NP_DNQ_**, not the free NR (e.g., NR fully stains cells with 10 min [55]), is the dominant factor of this fluorescence increase. Through the cellular uptake study, we confirmed that **NR-NP_DNQ_** efficiently entered into MCF-7 cells within 1 h.

Human umbilical vein endothelial cells (HUVECs) were used to evaluate the cytotoxicity. The viability of HUVECs was determined by MTT assay in corresponding culture medium after 24 h incubation with DTX, polymers and micelles. PEG-PLL-DNQ and PEG-PLL-Oct showed low cytotoxicity against HUVECs with the concentrations up to 0.5 mg/mL (>90% viability, Appendix A). Furthermore, the cytotoxicity of DTX, **NP_DNQ_** and **DTX-NP_DNQ_** showed that LC_50_ for DTX was 99 ng/mL and **DTX-NP_DNQ_** was 310 ng/mL in HUVECs (Figure 5a). The results show a lower cytotoxicity with **DTX-NP_DNQ_** than free drug **DTX**, against HUVECs, because **NP_DNQ_** strongly retains the hydrophobic aromatic DTX in the core of micelle due to hydrophobic and π–π stacking interactions.

### 3.4. Measurement of Light-Induced Destabilization of Micellar Nanoparticles

To investigate the light responsiveness of **DTX-NP_DNQ_**, the micellar nanoparticles were irradiated by 365 nm UV light, and DLS measurements were conducted to confirm the light-triggered destabilization process of **DTX-NP_DNQ_**. DLS results showed a change in the micellar diameter in response to UV light (Figure 5b). The irradiated **DTX-NP_DNQ_** (50 mW/cm^2^, 10 min) showed a decrease in its hydrodynamic size from 143 nm prior to irradiation to 18 nm and additionally, 63% of DTX was released into the solution during the light-triggered process; however, the DTX release profile after 1, 3, and 5 min of UV irradiation could not be obtained due to the destabilization of **DTX-NP_DNQ_** (i.e., partial IC from DNQ) during dialysis. These results revealed that the micelles dissociated partly under UV light, which was attributed to the Wolff rearrangement of hydrophobic DNQ moieties into hydrophilic IC molecules (Appendix A). The change in the hydrophilic–hydrophobic balance of **NP_DNQ_** then destroyed the structure of the micelles.

The light-triggered effect of the micelles was studied with MCF-7 cells (Figure 5c,d); Cells were incubated with **DTX-NP_DNQ_** for 2 h, washed with fresh media, irradiated by 365 nm UV light for 10 min (50 mW/cm^2^), and then further incubated for 24 h. Prior to irradiation, the cell viability of **DTX-NP_DNQ_** (400 ng/mL of DTX) was 55%. Upon irradiation with UV light, the cell viability was decreased to approximately 36%, similar to the cell viability of free DTX (25%), while there was no significant difference in the MCF-7 cell viability of **NP_DNQ_** with UV irradiation (Figure 5c). The study showed that **DTX-NP_DNQ_** was more cytotoxic than **NP_DNQ_** with UV irradiation because most of **DTX-NP_DNQ_** were disrupted by UV irradiation and then released DTX into MCF-7 cells.

## 4. Conclusions

In summary, we developed a photo-responsive micellar system with the DNQ-conjugated block copolymer. Hydrophobic aromatic DTX drugs were loaded into the micellar nanoparticles with enhanced stability and loading capacity, and then released into aqueous solution in a controlled manner under UV irradiation. Without light irradiation, the micellar nanoparticles were stable due to hydrophobic and π–π stacking interactions of DNQs with neighboring DNQs and DTX, making them suitable for biomedical applications within a living system. Investigations are underway on the potential use of these micellar nanoparticles for light-triggered intracellular delivery of hydrophobic aromatic anticancer drugs.

## Figures and Tables

**Figure 1 polymers-13-00377-f001:**
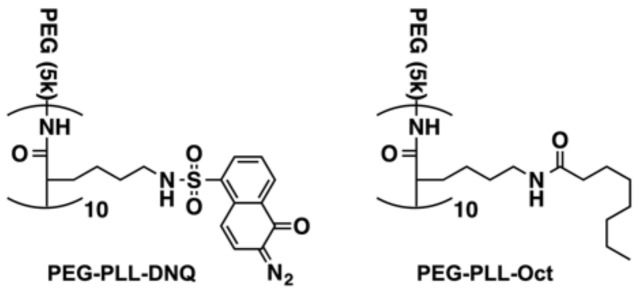
Structure of PEG-PLL-DNQ and PEG-PLL-Oct.

**Figure 2 polymers-13-00377-f002:**
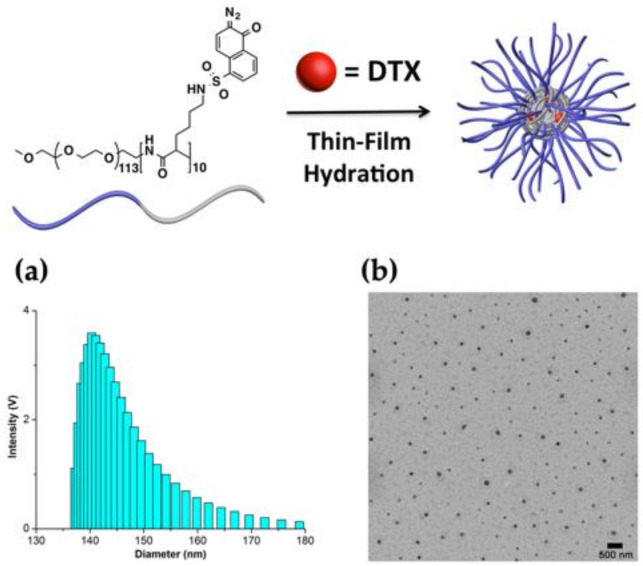
(**a**) DLS size distribution of **DTX-NP_DNQ_**; (**b**) TEM image of **DTX-NP_DNQ_** (scale bar = 500 nm).

**Figure 3 polymers-13-00377-f003:**
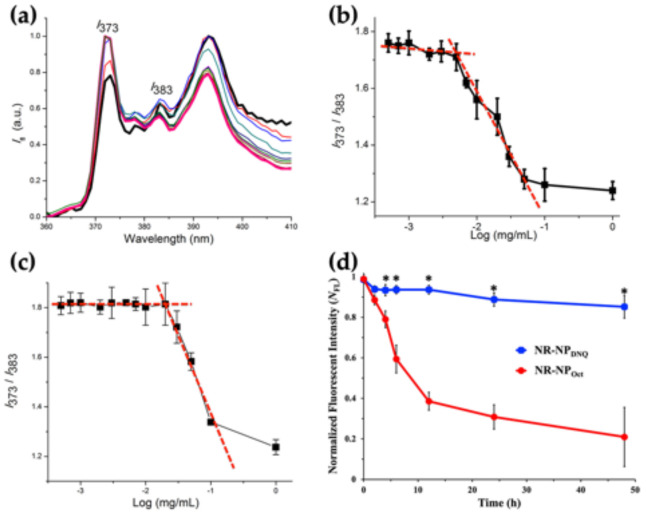
(**a**) Normalized emission spectra of pyrene in solutions of PEG-PLL-DNQ in PBS; emission intensity of pyrene at 373 and 383 nm versus the concentrations of (**b**) PEG-PLL-DNQ and (**c**) PEG-PLL-Oct; (**d**) stability of **NR-NP_DNQ_** (d-blue) and **NR-NP_Oct_** (d-red) in BSA solution. Data are means ± SD, *N* = 4, asterisks indicate *p* < 0.05.

**Figure 4 polymers-13-00377-f004:**
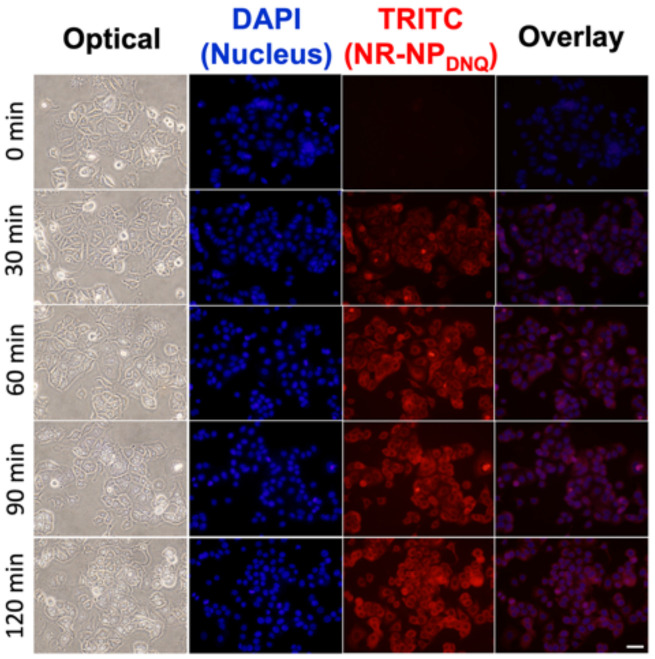
Fluorescence images of the internalization of **NR-NP_DNQ_** by MCF-7 cells for different time intervals: Nuclear staining with DAPI (blue color) and **NR-NP_DNQ_** (red color). (Scale bar: 20 μm).

**Figure 5 polymers-13-00377-f005:**
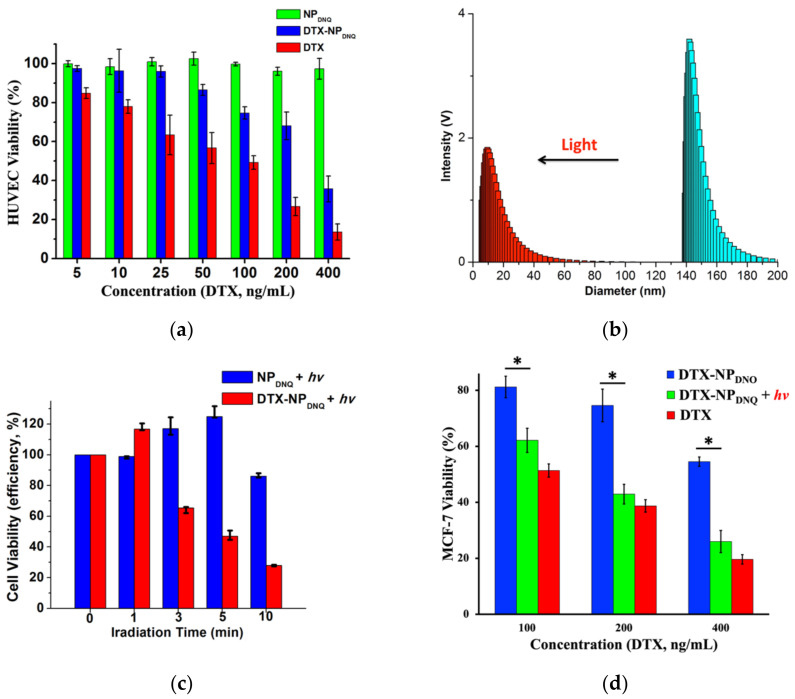
(**a**) Cytotoxicity of HUVECs with **NP_DNQ_** (green bar), **DTX-NP_DNQ_** (blue bar), and DTX (red bar); (**b**) DLS measurement of size change of **DTX-NP_DNQ_** upon UV light irradiation (365 nm, 50 mW/cm^2^, 10 min) in PBS; (**c**) MCF-7 Cells Viability with Light Irradiation (**NP_DNQ_** & **DTX-NP_DNQ_**); (**d**) Cytotoxicity of MCF-7 cells with different concentrations of **DTX-NP_DNQ_** (blue bar), **DTX-NP_DNQ_** with UV light irradiation (365 nm, 10 min, 50 mW/cm^2^) (green bar) and DTX (red bar). Data are means ± SD, *N* = 4, the concentrations of NPs were adjusted to the concentration of DTX by UV-Vis spectroscopy, asterisks indicate *p* < 0.05.

## Data Availability

Raw data are available from the corresponding author upon request.

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
