# Peer review of "Light-Responsive Polymeric Micellar Nanoparticles with Enhanced Formulation Stability"

_polymers, 2021, doi:10.3390/polym13030377_

Round 1
Reviewer 1 Report
Kim and coworkers developed light-responsive polymeric nanoparticles with enhanced micellar stability. This is a well written paper that should be published in Polymers after the issues are addressed.
- The polydispersity (PDI) of the polymer purchased or prepared should be added in the text.
- The image resolution of figure 3 should be improved to a publishing level.
- In figure 5, the authors should make the concentration clear, does it mean the concentration of DTX or DTX-NP?
- Several studies (Biomacromolecules 2012, 13 (3), 814-825; Colloids Surf. B 2014, 115, 384-390; Acta biomaterialia 2020, 110, 37-67) should be included in the refs [39-45] (line 49).
- ‘MW’ should be changed to ‘MW’ (line 74).
Author Response
Reviewer 1 :Kim and coworkers developed light-responsive polymeric nanoparticles with enhanced micellar stability. This is a well written paper that should be published in Polymers after the issues are addressed.
Thank you so much for your comments and suggestions. We have revised/corrected all what the reviewer 1 recommended.
- The polydispersity (PDI) of the polymer purchased or prepared should be added in the text.
Response: PDI that was provided from the Alamanda Polymers is added in the materials information (line 75-76).
- The image resolution of figure 3 should be improved to a publishing level.
Response: Now all figures including figure 3 are 600 dpi resolution.
- In figure 5, the authors should make the concentration clear, does it mean the concentration of DTX or DTX-NP?
Response: We added “the concentrations of NPs were adjusted to the concentration of DTX” on line 179 and 316-317.
- Several studies (Biomacromolecules 2012, 13 (3), 814-825; Colloids Surf. B 2014, 115, 384-390; Acta biomaterialia 2020, 110, 37-67) should be included in the refs [39-45] (line 49).
Response: As the reviewer recommended, these three refs. are included as refs. 46-48 (line 49 and 437-444)
- ‘MW’ should be changed to ‘MW’ (line 74).
Response: It is corrected (line 75)

Reviewer 2 Report
The manuscript described the obtention of block copolymer micelles, based on two types of block copolymers derived of PEG, in which one block is light-responsive, thus forming systems suitable for drug delivery applications, which is also tested in the current work. The text is short, but clear and concise and may be of interest to the readers of the special issue of Polymers. Based on that, I support its publication after some revisions, as stated below:
Major comment:
1. My main concern is related to the existence of true micelles in the systems (equilibrium structures, that is thermodynamically stable entities) or kinetically stable aggregates (like particles or vesicles). The experiment described in section 1.7 could solve this question, since micelles have the same properties independently of the path they are prepared, but the results are not presented in the manuscript (!). Please, show these results and, based on that, evaluate if the term "micelle" is the best to be employed in this case. Information on the visual aspect of the samples may also help in this issue (micellar solutions are transparent while other aggregates are turbid or milky).
Minor comments:
1. The way that the number of repeating units, or the molecular weight (MW), of each block of the copolymers should be standardized. For the mPEG block, the subscribed symbol (5k) seems to indicate the MW of this block, while for PLKF the subscribed number (10) seems to indicate the number of monomer repeating units. Please, clarify that.
2. The (poli)dispersity index of the initial block copolymer should also be informed.
3. If possible, the chemical structures of DTX and Nile Red should be presented in the manuscript.
4. The drying procedure employed in TEM analysis may cause considerable changes in the size and shape of the micelles since they only exist in solution. More appropriate imaging should be performed with negative staining TEM or cryo-TEM. Please, make sure that readers will be aware that the obtained TEM images may not represent the "true" state of the micelles (or any other kind of aggregate) in the original solution.
5. No lipid membrane is formed in the systems, but block copolymer micelles. Please, correct that (page 3, line 115). In this sense, what is the reason for using such a procedure to prepare the block copolymer micelles (item 1.5)? The thin-film-hydration method is largely used to prepare lipid vesicles (or liposomes) but rarely employed to prepare micelles in aqueous solutions. Linked to my first comment, such a preparation procedure suggests me that the samples may be formed by dispersions/suspensions and not true micellar solutions. Again, this issue has to be better addressed.
6. The preparation of PEG-PLL-Oct micelles loaded with DTX or Nile Red, as well as their CMC determination, are not reported in any part of the manuscript. It was supposed to appear in sections 1.5 and 1.6. Please, add that.
7. The term "micellar suspension" should be avoided if the system is really comprised of a true micellar solution (relating to my first concern - above)
8. HPLC is the separation technique, not the detection technique. Please, specify which kind of detector (UV, Vis, Refractive Index, etc.) was coupled to HPLC to determine the DTX concentration.
9. How was obtained the data on the amount of DTX released by DTX-NP_DNQ after UV irradiation (page 7, line 296)? I suppose it is like the procedure described in topic 1.10, but it is not clear.
10. The concentrations displayed in x-axis of Figure 5a and b are related to which component (DTX?)? To make data comparable, all the different samples (micelle, DTX and micelle + DTX) should have to contain the same amount of the drug in each case. Please, specify this.
11. In Figure S5, which sample is PLL-IC? Such a sample with this label was not mentioned in any part of the text.
12. In the references, some journals have their names abbreviated, while others are presented in full. Please, check the format required by the journal and standardize it.
Author Response
Reviewer 2 :The manuscript described the obtention of block copolymer micelles, based on two types of block copolymers derived of PEG, in which one block is light-responsive, thus forming systems suitable for drug delivery applications, which is also tested in the current work. The text is short, but clear and concise and may be of interest to the readers of the special issue of Polymers. Based on that, I support its publication after some revisions, as stated below:
Thank you so much for your comments and suggestions. We have revised/corrected all what the reviewer 1 recommended.
Major comment:
- My main concern is related to the existence of true micelles in the systems (equilibrium structures, that is thermodynamically stable entities) or kinetically stable aggregates (like particles or vesicles). The experiment described in section 1.7 could solve this question, since micelles have the same properties independently of the path they are prepared, but the results are not presented in the manuscript (!). Please, show these results and, based on that, evaluate if the term "micelle" is the best to be employed in this case. Information on the visual aspect of the samples may also help in this issue (micellar solutions are transparent while other aggregates are turbid or milky).
Response: As the reviewer commented, one of major concerns about micelle systems is the stability. That is the core concept of this project and why we showed the results of section 1.7 and 1.8, but somehow we missed these very important comments in the manuscript (especially thanks to the reviewer and we added these in the manuscript): “Subsequently, the solution was dialyzed in water for 24 h (molecular weight cutoff: 3.5 kDa)” on line 117-118 and “The reconstituted micellar nanoparticles were used for all data” on line 138. The reviewer asked us to provide the results but all the data in the manuscript were obtained from the reconstituted micellar nanoparticles; because as shown in Figure 3d, the micellar nanoparticles will eventually lose their stability if they are stored in a solution like all other micelles. Thus, the reviewer’s major concern is resolved.
Minor comments:
- The way that the number of repeating units, or the molecular weight (MW), of each block of the copolymers should be standardized. For the mPEG block, the subscribed symbol (5k) seems to indicate the MW of this block, while for PLKF the subscribed number (10) seems to indicate the number of monomer repeating units. Please, clarify that.
Response: We added the clarification: “5k PEG conjugated with 10 lysine units” on line 75 and “5k PEG conjugated with 10 lysine units” on line 197-198.
- The (poli)dispersity index of the initial block copolymer should also be informed.
Response: PDI that was provided from the Alamanda Polymers is added in the materials information (line 75-76).
- If possible, the chemical structures of DTX and Nile Red should be presented in the manuscript.
Response: Unfortunately, we are not able to add an additional figure into the manuscript, but we added the chemical structures in Figure S1.
- The drying procedure employed in TEM analysis may cause considerable changes in the size and shape of the micelles since they only exist in solution. More appropriate imaging should be performed with negative staining TEM or cryo-TEM. Please, make sure that readers will be aware that the obtained TEM images may not represent the "true" state of the micelles (or any other kind of aggregate) in the original solution.
Response: I appreciate the reviewer’s details but ‘drying samples reduce the size of micelles in TEM’ is well-known in the field. There is no need to further explain this to the readers.
- No lipid membrane is formed in the systems, but block copolymer micelles. Please, correct that (page 3, line 115). In this sense, what is the reason for using such a procedure to prepare the block copolymer micelles (item 1.5)? The thin-film-hydration method is largely used to prepare lipid vesicles (or liposomes) but rarely employed to prepare micelles in aqueous solutions. Linked to my first comment, such a preparation procedure suggests me that the samples may be formed by dispersions/suspensions and not true micellar solutions. Again, this issue has to be better addressed.
Response: We corrected it from ‘the drug-containing lipid membrane was formed’ to ‘the drug-containing block copolymer film was formed’ (line 116). Also, we added the missing step, “Subsequently, the solution was dialyzed in water for 24 h (molecular weight cutoff: 3.5 kDa)” on line 117-118.
- The preparation of PEG-PLL-Oct micelles loaded with DTX or Nile Red, as well as their CMC determination, are not reported in any part of the manuscript. It was supposed to appear in sections 1.5 and 1.6. Please, add that.
Response: we added “The similar procedures were used to formulate PEG-PLL-Oct-based micellar nanoparticles” on line 119-120.
- The term "micellar suspension" should be avoided if the system is really comprised of a true micellar solution (relating to my first concern - above)
Response: We added missing comments and procedures in the manuscript: i) “The reconstituted micellar nanoparticles were used for all data” (line 138); and ii) the solution was dialyzed in water for 24 h (molecular weight cutoff: 3.5 kDa) (line 118).
- HPLC is the separation technique, not the detection technique. Please, specify which kind of detector (UV, Vis, Refractive Index, etc.) was coupled to HPLC to determine the DTX concentration.
Response: As provided in section 1.2, we used a Hewlett Packard/Agilent series 1100 (Agilent, Santa Clara, CA) equipped with an analytical C18 reverse phase column (Kinetex, 75 × 4.6 mm, 2.6 μ, Phenomenex, Torrance, CA) and it includes a UV-vis detector. HPLC has been widely used for LC and LE (Biomacromolecules 2019, 20, 1545−1554) in the field and so there is no need to further explain this.
- How was obtained the data on the amount of DTX released by DTX-NP_DNQ after UV irradiation (page 7, line 296)? I suppose it is like the procedure described in topic 1.10, but it is not clear.
Response: In drug delivery systems (DDS), the section 1.10 and HPLC for LC/LE are common methods in the field. I appreciate the reviewer’s comment, but more detailed explanations for these procedures are not necessary to the readers.
- The concentrations displayed in x-axis of Figure 5a and b are related to which component (DTX?)? To make data comparable, all the different samples (micelle, DTX and micelle + DTX) should have to contain the same amount of the drug in each case. Please, specify this.
Response: We corrected Figures 5a and 5b from “concentration (ng/mL)” to “concentration (DTX, ng/mL)” and added ‘the concentrations of NPs were adjusted to the concentration of DTX’ on line 316-317.
- In Figure S5, which sample is PLL-IC? Such a sample with this label was not mentioned in any part of the text.
Response: We corrected Figure 5S from “PLL-IC” to “PEG-PLL-Oct”.
- In the references, some journals have their names abbreviated, while others are presented in full. Please, check the format required by the journal and standardize it.
Response: We corrected Endnote issues.

Reviewer 3 Report
Lee and coworkers reported a PEG-Poly-lysine block copolymer that was functionalized with DNQ moieties. These DNQ moieties in theory can be cleaved by UV radiation and result in the release of the encapsulated drug/dye. Several issues shall be addressed before further consideration.
- Figure 2a, please plot the x-axis for DLS data in log scale.
- Line 253, the authors hypothesized that the pi-pi stacking could be a reason for the DNQ polymers having higher LE and LC of DTX than the Oct polymers. Another factor that possibly affect the LE and LC is the logP value of polymers and drugs/dyes. Several articles (DOI: 10.1016/j.ijpharm.2010.06.008; DOI: 10.1021/acs.macromol.5b01758; DOI: 10.1021/acs.macromol.9b02595) has found that the logP of dyes also affects the their LE and LC value in polymeric micelles. The authors are suggested to test the LE and LC of Nile red plus another dye having a different logP.
- Figure 4, if the authors tested the cellular uptake of DNQ polymers loaded with Nile Red, please clearly state it in the Methods and figure caption. Scale bars are missing from the figures too.
- Line 272, the authors stated that "As shown in Figure 3d, less than 7 % of hydrophobic NR was released from the micelle within 6 h." It is not quite a direct correlation between stability test in the presence of serum and the dye release behavior in terms of cells (complications in lipid bilayer, membrane proteins, cell surface glycome). As the authors stated in the manuscript, the fluorescence of Nile red turns on in hydrophobic environment. The lipid membrane in cells provides such a big difference compared to serum proteins only (and in PBS). The authors are required to add a comparison panel with Nile red only (dilute from DMSO stock), or switch the cargo from Nile red to doxorubicin.
- The UV-triggered cleavage feature of PEG-PLL-DNQ should be further explained by a mechanism demonstration in Figure 1.
- What are the DLS data and dye loading profiles of these PEG-PLL-DNQ micelles after UV irradiation? Please provide the data.
- Figure 5b, the cell viability data for UV-irradiation only group is missing.
Author Response
Reviewer 3 :Lee and coworkers reported a PEG-Poly-lysine block copolymer that was functionalized with DNQ moieties. These DNQ moieties in theory can be cleaved by UV radiation and result in the release of the encapsulated drug/dye. Several issues shall be addressed before further consideration.
Thank you so much for your comments and suggestions. We have revised/corrected all what the reviewer 1 recommended.
- Figure 2a, please plot the x-axis for DLS data in log scale.
Response: Thank you for your suggestion and we agree that it will show a better image, but as you can see Figure S6, we really want to show the detailed size change of the micellar nanoparticles before/after irradiation. Thus, I would like to keep the current image because the readers will clearly see the difference.
- Line 253, the authors hypothesized that the pi-pi stacking could be a reason for the DNQ polymers having higher LE and LC of DTX than the Oct polymers. Another factor that possibly affect the LE and LC is the logP value of polymers and drugs/dyes. Several articles (DOI: 10.1016/j.ijpharm.2010.06.008; DOI: 10.1021/acs.macromol.5b01758; DOI: 10.1021/acs.macromol.9b02595) has found that the logP of dyes also affects the their LE and LC value in polymeric micelles. The authors are suggested to test the LE and LC of Nile red plus another dye having a different logP.
Response: I appreciate the suggested articles. The logP values of 1,2-naphthoquinone (i.e., not diazonaphthoquinone) and octane are 2.21 and 4.78, respectively. This means that NPOct should have higher LE and LC with DTX than those of NPDNQ, but the results are opposite. This clearly indicates that the pi-pi stacking interactions contribute high LE and LC of NPDNQ, but not hydrophobicity (i.e., logP).
- Figure 4, if the authors tested the cellular uptake of DNQ polymers loaded with Nile Red, please clearly state it in the Methods and figure caption. Scale bars are missing from the figures too.
Response: We added ‘20x magnification’ in the Figure 4 caption. (line 291-292) and formulation of NP with NR is stated in section 1.5.
- Line 272, the authors stated that "As shown in Figure 3d, less than 7 % of hydrophobic NR was released from the micelle within 6 h." It is not quite a direct correlation between stability test in the presence of serum and the dye release behavior in terms of cells (complications in lipid bilayer, membrane proteins, cell surface glycome). As the authors stated in the manuscript, the fluorescence of Nile red turns on in hydrophobic environment. The lipid membrane in cells provides such a big difference compared to serum proteins only (and in PBS). The authors are required to add a comparison panel with Nile red only (dilute from DMSO stock), or switch the cargo from Nile red to doxorubicin.
Response: We agree with the reviewer’s comment, “It is not quite a direct correlation between stability test in the presence of serum and the dye release behavior in terms of cells (complications in lipid bilayer, membrane proteins, cell surface glycome)”. Thus, we delete “As shown in Figure 3d, less than 7 % of hydrophobic NR was released from the micelle within 6 h”, but the additional experiment, a comparison panel with Nile red only, is not necessary because many articles have shown that NR will stain the cells immediately (i.e., it will show the saturation of fluorescence intensity at 0 min).
- The UV-triggered cleavage feature of PEG-PLL-DNQ should be further explained by a mechanism demonstration in Figure 1.
Response: The image in the abstract clearly shows the mechanism (please see below) but we added a more detailed mechanism in Figure S2 as the reviewer recommended.
- What are the DLS data and dye loading profiles of these PEG-PLL-DNQ micelles after UV irradiation? Please provide the data.
Response: The data are in Figure S6.
- Figure 5b, the cell viability data for UV-irradiation only group is missing.
Response: The data are in Figure S7.

Round 2
Reviewer 3 Report
The authors mentioned that "We have revised/corrected all what the reviewer 1 recommended". Indeed, several questions from Reviewer 2 and 3 were not properly addressed.
- "Unfortunately, we are not able to add an additional figure into the manuscript, but we added the chemical structures in Figure S1." The authors still can add the content of Figure S1 into Figure 1.
- Similarly, Figure S6 and S7 are important information to be shown in the main manuscript, not the SI. Figure S6 is missing the dye release profile from the micelles after UV irradiation. The authors are strongly recommended to provide this missing key information, and combine them together with Figure S6 into Figure 3. Figure S7 also has important contents that belong to Figure 5.
- "I appreciate the reviewer’s details but ‘drying samples reduce the size of micelles in TEM’ is well-known in the field. There is no need to further explain this to the readers." This would indicate that there are plenty of references addressing the issues of drying samples for TEM, or at least pointing it out. Please provide this rationale in the manuscript with references.
- Examples from the authors' response: "HPLC has been widely used for LC and LE (Biomacromolecules 2019, 20, 1545−1554) in the field and so there is no need to further explain this." "In drug delivery systems (DDS), the section 1.10 and HPLC for LC/LE are common methods in the field. I appreciate the reviewer’s comment, but more detailed explanations for these procedures are not necessary to the readers." It is hard to agree that the authors believe that providing detailed procedures is not necessary. On the contrary, providing details is one of the reasons that research can move forward from everyone's mutual efforts. Please, provide the details.
- In Figure 5, after the authors adjusted the denotations. In the latest figure, does the DTX concentration represent the amount of DTX initially being loaded into the micelles, or the HPLC characterization results after reconstituting the micelles? Please, elaborate.
- "PDI that was provided from the Alamanda Polymers is added in the materials information (line 75-76)." "PDI" is not recommended by IUPAC (DOI: 10.1351/PAC-REC-08-05-02). Please, correct it to "dispersity (Đ)".
- The logP analysis from the authors are: "The logP values of 1,2-naphthoquinone (i.e., not diazonaphthoquinone) and octane are 2.21 and 4.78, respectively. This means that NPOct should have higher LE and LC with DTX than those of NPDNQ, but the results are opposite. This clearly indicates that the pi-pi stacking interactions contribute high LE and LC of NPDNQ, but not hydrophobicity (i.e., logP)." This is a good point however missing the experimental evidence. The authors did not provide the LE and LC data for Nile red, which is quite important as they later tested the cellular uptake with Nile red. Without such variations (barely minimal) in cargo hydrophobicity, the authors cannot exclude that logP does not have any effect, at least at this point. Please, provide the data, take it along for the logP factor analysis, and include in the manuscript.
- "but the additional experiment, a comparison panel with Nile red only, is not necessary because many articles have shown that NR will stain the cells immediately (i.e., it will show the saturation of fluorescence intensity at 0 min)." Could the authors provide the information for the readers because it may be misleading if the manuscript does not point it out? Such an evidence do need to have either experimental proof or references to support.
Round 3
Reviewer 3 Report
- The authors explained that "For the dye (drug) release profile, we were not able to obtain reliable data from 3 and 5 mins irradiation because, as described in section 1.10, we determined the amount of DTX using the dialysis method, which means it takes a day and the unstable nanoparticles slowly burst inside the dialysis tube (more IC from DNQ). " It is reasonable therefore should be included in the manuscript discussion.
- The response from the authors is still missing references to support their following claim: "...many articles have shown that NR will stain the cells immediately (i.e., it will show the saturation of fluorescence intensity at 0 min)." Again, either provide the Nile red only control experiment in Figure 4, or provide references to explain why this important control experiment is excluded from the study.
- Figure 4 needs to be labeled with scale bars, not magnification.
